# Elucidating the role of human skeletal muscles in the pathogenesis of enterovirus D68 infection

Brigitta M Laksono[1], Atze J Bergsma[2,3,4,*], Alessandro Iuliano[2,3,4,*], Dominique Y Veldhoen[1], Stefan van Nieuwkoop[1], Marjan Boter[1], Lonneke Leijten[1], Lisa Bauer[1], Bas B Oude Munnink[1], WWM Pim Pijnappel[2,3,4], Debby van Riel[1]

Enterovirus D68 (EV-D68) is an emerging respiratory virus associated with extra-respiratory complications, especially acute flaccid myelitis. However, the pathogenesis of acute flaccid myelitis is not fully understood. It is hypothesised that through infection of skeletal muscles, the virus further infects motor neurons via the neuromuscular junction. We hypothesise that EV-D68 infection of human skeletal muscles can impair muscle function directly, thereby contributing to the development of EV-D68–associated muscle weakness. Here, we inoculated human induced pluripotent stem cell–derived skeletal muscle myotubes grown in 2D and 3D with different EV-D68 isolates, which resulted in a productive infection and cell death. We showed through neuraminidase treatment that sialic acids facilitate infection of these cells. EV-D68 infection of the 3D model led to tissue damage, reduction of contractile force, and hampered muscle regeneration. Altogether, we showed that human skeletal muscle can act as an extra-respiratory replication site and infection of skeletal muscles may contribute to EV-D68–associated muscle weakness.

## Introduction

First isolated in 1962, enterovirus D68 (EV-D68) was initially only associated with mild respiratory disease (1). However, during the global EV-D68 outbreaks in 2014, the virus became associated with severe respiratory disease and extra-respiratory complications, especially neurological ones. Since then, the virus has caused biennial outbreaks and the number of EV-D68–confirmed cases increased (2). The implementation of COVID-19 measures during the pandemic helped breaking this pattern, but once the measures were lifted, the number of confirmed cases increased again (3, 4, 5, 6).

EV-D68 is currently categorised into three clades: A, B, and C. Clade A is further subdivided into subclades A1 and A2 (previously known as clade D). Clade B is subdivided into subclades B1, B2, and B3. Viruses from all clades have been co-circulating before 2014. Currently, only viruses from subclades A2 and B3 are circulating in North America, Europe, Australia, Africa, and Asia (7). However, because regular surveillance is not commonly conducted in all countries, it is unknown whether these clades truly represent the currently circulating viruses worldwide.

Of all EV-D68–associated extra-respiratory complications, acute flaccid myelitis (AFM) is reported most frequently. The early manifestations of AFM include headache; facial or eyelid droop; slurred speech or difficulty swallowing; pain in the neck, back, or limbs; sudden arm or leg weakness; and hyporeflexia (8). Autopsy showed that EV-D68 infects motor neurons in the anterior horn of the spinal cord, which supports the view that direct infection of spinal cord motor neurons can cause AFM (9). Clinical outcome of AFM differs widely, ranging from the more common lifelong muscle weakness and atrophy to the very rare cases of complete recovery. Within weeks to months after the onset of paralysis, nearly all patients had muscle atrophy in the affected limbs, diffused muscle aches, and limb pain (10). It is currently unclear what factors determine these different outcomes.

The pathogenesis of EV-D68–associated AFM is still poorly understood and therefore often extrapolated from the pathogenesis of poliomyelitis. From the primary replication site, which is the respiratory tract, EV-D68 is considered to spread via the haematogenous route into other organs, including the spinal cord. Like in poliomyelitis, the virus may infect skeletal muscles, where it may spread further to motor neurons via the neuromuscular junction. Studies in interferon (IFN)–α/ß receptor-deficient (11, 12) and neonatal (13, 14) mice have shown that EV-D68 infects skeletal muscles and motor neurons, the latter ultimately leading to paralysis. Interestingly, in one study, after an intranasal inoculation, paralysis was observed in the presence of myositis and myofibre degeneration in the absence of EV-D68 infection of motor neurons,

[1]Department of Viroscience, Erasmus MC, Rotterdam, The Netherlands  [2]Department of Paediatrics, Erasmus MC, Rotterdam, The Netherlands  [3]Department of Clinical Genetics, Erasmus MC, Rotterdam, The Netherlands  [4]Centre for Lysosomal and Metabolic Diseases, Erasmus MC, Rotterdam, The Netherlands

Correspondence: d.vanriel@erasmusmc.nl
*Atze J Bergsma and Alessandro Iuliano contributed equally to this work

suggesting that muscle weakness may also occur without neurological involvement (11). Direct infection of skeletal muscles, and the associated inflammatory responses and cell death, may thus play a role in EV-D68–associated limb weakness. In this study, we aim to understand the direct role of human skeletal muscles in the pathogenesis of EV-D68 infection by assessing the susceptibility and permissiveness of human induced pluripotent stem cell (hiPSC)–derived myotubes to infection of EV-D68. We included strains from clades A and B isolated before and after 2014, some of which are known to be associated with AFM (15) or can cause paralytic disease in mice (16). We also investigated whether EV-D68 infection in 3D tissue-engineered skeletal muscles (3D TESMs) results in muscle weakness.

# Results

## Whole-genome sequencing of EV-D68/A1, EV-D68/A2, and EV-D68/B2 stocks

To ensure that EV-D68/A1, EV-D68/A2, and EV-D68/B2 stocks that were used in this study were genetically similar to the clinical isolates from which they derived, we generated near-whole-genome sequences for all isolates and investigated whether cell culture–adaptive amino acid substitutions were acquired upon virus passage. The previously described cell culture–adaptive amino acid substitution at position 271 of VP1, which allows EV-D68 to use heparan sulphate as an additional receptor and thus influence the tropism of the virus, was not present in any of the virus stocks (17). In EV-D68/A1 stock, we did not detect any amino acid substitutions in other positions (Table 1). In EV-D68/A2 stock, we only detected one amino acid substitution in the VP1 region (T132K), which has never been reported previously. This substitution is located in a highly variable structure in the DE loop. In EV-D68/B2 stock, amino acid substitutions were observed in VP2 (P56T and H98Y), VP3 (V166I), and VP1 (D285Y); the last one has never been reported previously (Table 1 and Fig S1). The substitutions in the EV-D68/B2 stock were mapped on the capsid structure of EV-D68 in Fig S1A, in which we showed that the substitution in VP1 was close to the sialic acid binding site (Fig S1B), but does not seem to be involved in sialic acid binding (Fig S1C). The substitutions in the VP2 region were located at a VP2-VP2 interface (Fig S1D). Whether the observed substitutions have an influence on the phenotypic characteristics, receptor binding, or replication is currently not understood.

## HiPSC-derived 2D myotubes are susceptible and permissive to EV-D68 infection

To assess whether human skeletal muscles are susceptible to infection of EV-D68 of different subclades, we inoculated hiPSC-derived 2D myotubes from three donors with EV-D68/A1 and EV-D68/A2 at an MOI of 0.01, 0.1, and 1, and monitored the progression of the infection daily up to 72 h post-inoculation (hpi). Infected cells were detected at 24 hpi (Figs 1A and S2) in EV-D68/A1- and EV-D68/A2-inoculated myotubes, and the infection progressed over time, as characterised by the appearance of a cytopathic effect (CPE), which

**Table 1.** Amino acid substitutions present in EV-D68/A1, EV-D68/A2, and EV-D68/B2 stocks included in this study.

| Isolate | Amino acid at position in protein | | |
|---------|------|------|------|
| | VP2 | VP3 | VP1 |
| A1 | — | — | — |
| A2 | — | — | T132K |
| B2 | P56T<br>H98Y | V166I | D285Y |

was apparent as rounding, detachment, and eventually cell death, as well as the increased number of infected cells. Mock-inoculated 2D myotubes typically have a long, multinucleated phenotype and can sometimes have a syncytia-like appearance. Upon infection with EV-D68, the cells became rounder around the nucleus area and thinner along the tube, with a gradual decrease in myosin expression, and the cells were subsequently detached (Fig S3). Overall, CPE was more prominent in cells inoculated at MOI 1 than at MOI 0.1 and 0.01 (Fig S2). In addition, at MOI 1, CPE was more prominent in EV-D68/A1-inoculated myotubes than in EV-D68/A2-inoculated ones at 24 hpi.

Because we observed that EV-D68 isolates from clade A can infect and replicate efficiently in 2D myotubes, we investigated whether viruses from clade B also have similar myotropism and myovirulence. We inoculated myotubes derived from different donors with EV-D68/B2 at MOI 0.1. The inoculation with EV-D68/B2 resulted in progression of CPE over time and increased number of infected cells, similar to what we observed in EV-D68/A-inoculated myotubes (Fig 1A). We observed that the progression of EV-D68/B2 CPE, signified by the increased number of dead and infected cells at 24 hpi, resembled that of EV-D68/A1 instead of EV-D68/A2.

To further investigate whether the infection of 2D myotubes resulted in a productive infection, we measured the intracellular viral RNA levels and infectious virus titres in the supernatants (Fig 1B and C). The intracellular EV-D68/A1, EV-D68/A2, and EV-D68/B2 RNA levels significantly increased and reached a plateau at 24 hpi in almost all cells inoculated at MOI 0.1 (Fig 1B). In myotubes inoculated with EV-D68/A1 and EV-D68/A2 at a lower or higher MOI, the increase was also observed and the plateau was reached similarly at 24 hpi (Fig S4A).

Accordingly, EV-D68/A1, EV-D68/A2, and EV-D68/B2 titres in the supernatants increased over time for all inoculation conditions (Figs 1C and S4B). Between 0 and 24 hpi, there was a statistically significant increase in virus titres in all donors inoculated with EV-D68/A1 and EV-D68/A2, but not EV-D68/B2. When we investigated the donor-to-donor difference in viral titres, we observed significantly higher EV-D68/A1 titre in Donor 1 than in Donors 2 and 3 at 72 hpi (MOI 0.1) or at 24 and 48 hpi (MOI 0.01). EV-D68/A2 titres were significantly higher at 0 hpi (MOI 0.1) and 48 hpi (MOI 0.01) in Donor 1 than in Donors 2 and 3.

## Infection of hiPSC-derived 2D myotubes was largely mediated by α2,3- and α2,6-linked SAs

EV-D68 can bind to α2,3- and α2,6-linked SAs to initiate entry in target cells (18, 19, 20). To investigate whether α2,3- and α2,6-linked

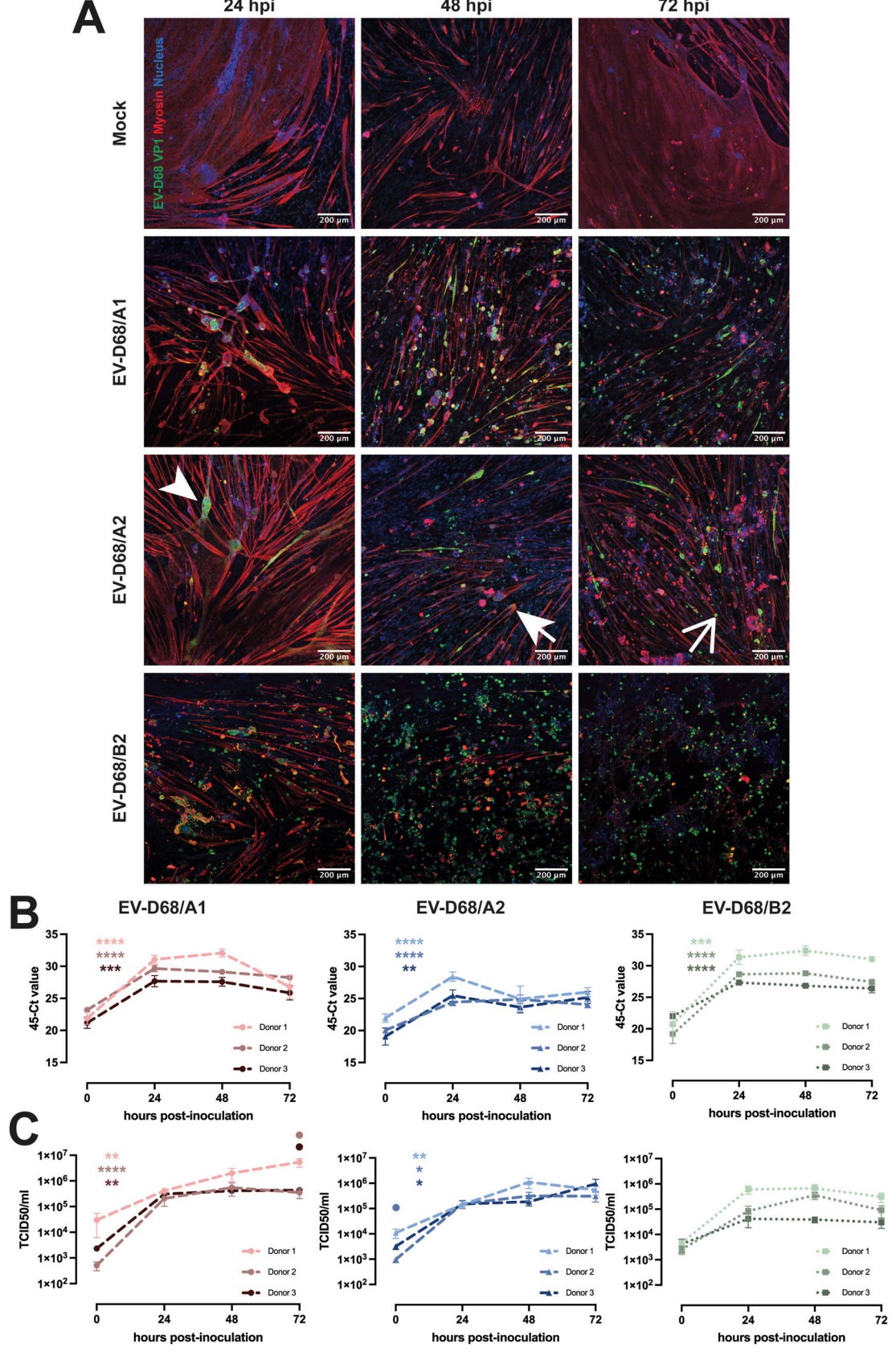

SAs mediate EV-D68 infection on hiPSC-derived 2D myotubes, we removed cell surface SAs with *Arthrobacter ureafaciens* neuraminidase (ANA) before inoculation with EV-D68/A1 or EV-D68/A2 at MOI 0.1. ANA treatment resulted in a visible decrease of $\alpha 2,6$-linked SA, although this was less prominent for $\alpha 2,3$-linked SA (Fig 2A). The treatment also resulted in a reduction of VP1[+] cells in EV-D68–inoculated myotubes (Fig 2B) and a significant decrease of intracellular viral RNA levels (Fig 2C) and virus titres in the supernatant at 24 hpi (Fig 2D).

### Inoculation of hiPSC-derived 3D TESMs with EV-D68 resulted in infection and reduced muscle function

To investigate whether EV-D68 infection of human skeletal muscles can result in reduction or loss of muscle contractile force, we inoculate hiPSC-derived 3D TESMs with EV-D68/A1 or EV-D68/A2 at MOI 0.1 and measured the contractility of the 3D TESMs at 2 and 7 days post-inoculation (dpi) (Fig 3A). Inoculation of 3D TESMs resulted in a productive infection, as marked by an increase of EV-D68/A1 and EV-D68/A2 titres in the supernatants over time (Fig 3B), with the titre of EV-D68/A2 becoming significantly higher than EV-D68/A1 during the peak of infection at 4 dpi. Twitch and tetanic contractile forces of the EV-D68–inoculated TESMs were measured by stimulating the tissues with a frequency of 1 and 20 Hz, respectively, at 2 and 7 dpi. Briefly, electrical stimulation was applied onto the TESMs and pillar displacement was tracked with high-speed video imaging as previously described (21). A functional TESM will show a single twitch contraction when stimulated with 1 Hz and reach a maximum tetanic contraction with a 20-Hz stimulus. These contractile forces can then be plotted into a graph (in mN). At 2 dpi, we observed a significant decrease of twitch contractile force in EV-D68/A1- (average: 0.22 ± 0.05 mN) and EV-D68/A2-inoculated (average: 0.22 ± 0.04 mN) 3D TESMs compared with mock-inoculated 3D TESMs (average: 0.48 ± 0.10 mN). We also observed a trend towards a decrease of tetanic contractile force in EV-D68/A1- (average: 0.52 ± 0.12 mN) and EV-D68/A2-inoculated (average: 0.69 ± 0.13 mN) 3D TESMs compared with mock-inoculated ones (average: 1.15 ± 0.28 mN), but this decrease was not statistically significant (Fig 3C). At 7 dpi, the twitch and tetanic contractile forces of the mock-inoculated 3D TESMs increased compared with those at 2 dpi (average: 3.56 ± 0.81 mN and 8.57 ± 2.45 mN, respectively). In contrast, at 7 dpi, EV-D68/A1- and EV-D68/A2-inoculated 3D TESMs have completely lost their contractility (Fig 3D).

Infection of skeletal muscle cells in 3D TESMs was confirmed by detection of EV-D68 VP1[+] cells at 2 and 7 dpi (Figs 4 and S5). The infection progressed over time, resulting in destruction of the 3D TESM structure, as marked by the loss of titin[+] skeletal muscle cells

at 7 dpi (Fig 4). Pax7[+] satellite cells, which can differentiate into skeletal muscle cells and play an important role in muscle repair, were abundantly present in mock-inoculated tissues (2 and 7 dpi) and EV-D68–inoculated tissues (2 dpi), but were depleted at 7 dpi in EV-D68–inoculated 3D TESMs (Figs S6, S7, and S8). However, no infected satellite cells were detected in virus-inoculated 3D TESMs at any time points, suggesting that the depletion was not the results of a direct infection. In vivo, after a skeletal muscle injury, satellite cells will start to proliferate and enter the early regenerative stage (22). To investigate whether EV-D68 infection leads to proliferation of satellite cells, we investigated the presence of Ki67[+] proliferating cells in EV-D68–inoculated 3D TESMs. We observed that the presence of Ki67[+] cells in EV-D68/A1- and EV-D68/A2-inoculated 3D TESMs was reduced compared with the mock-inoculated 3D TESMs (Figs S6, S7, and S8). The reduction of Ki67[+] Pax7[+] cells suggested that there was no activation and proliferation of satellite cells after EV-D68 infection of 3D TESMs.

## Discussion

In this study, we showed that EV-D68 from subclades A1 (isolated in 2012), A2 (isolated in 2018), and B2 (isolated in 2012) replicated efficiently in hiPSC-derived 2D and 3D skeletal muscle models. Infection of these human skeletal muscle models was largely mediated by SAs. Lastly, we showed that EV-D68 infection of hiPSC-derived 3D TESMs results in loss of muscle contractile force that was accompanied by loss of muscle fibres and satellite cells over time.

The viruses included in this study did not contain the previously described cell culture–adaptive amino acid substitutions that affect the receptor binding preference (17, 23). We did observe other cell culture–adaptive amino acid substitutions in our EV-D68/A2 and EV-D68/B2 stocks. Some of these substitutions, except for those in the VP1 region, have been reported in previously and currently circulating strains in Nextstrain (24). P56T has been reported in one strain (accession code OP321151) belonging to subclade B3. H98Y occurred more often in EV-D68 strains from clade A, especially the currently circulating strains from subclade A2. 166I is naturally occurring in EV-D68 strains from clade B, whereas most strains from clade A have 166V. V166I has been detected in four strains belonging to clade A (accession codes KX255360, KM892500, KY767821, and KT803588). Although the physiological relevance of these amino acid substitutions is not understood, our findings highlight that enteroviruses adapt fast in cell culture (17, 25). We therefore recommend to sequence virus stocks before experimental use as cell culture adaptation may significantly alter the phenotypic characteristics of the viruses.

**Figure 1. EV-D68 infection of hiPSC-derived 2D myotubes.**
**(A)** Representative images of myotubes inoculated with EV-D68/A1, EV-D68/A2, and EV-D68/B2 at MOI 0.1, shown at 24, 48, and 72 hpi. Green: EV-D68 VP1; red: myosin; blue: nucleus. The arrowhead indicates a rounding cell; the thick arrow indicates a detached cell; the thin arrow indicates a dead cell. **(B, C)** Myotubes (n = 3 donors) were inoculated with EV-D68/A1, EV-D68/A2, and EV-D68/B2 at MOI 0.1. **(B, C)** Cells and supernatants were collected at 0, 24, 48, and 72 hpi for detection of (B) intracellular RNA and (C) infectious virus particles in the supernatant. Per donor, three experiments were performed with two technical replicates. Error bars indicate SEM. The asterisk indicates a significant difference between 0 and 24 hpi of each donor. The circle indicates a significantly higher virus titres in Donor 1 compared with Donors 2 (light red or blue) and 3 (dark red or blue) at a certain time point. Statistical analysis was performed using an unpaired *t* test (0 versus 24 hpi differences) and one-way ANOVA with multiple comparisons (donor-to-donor differences). *P < 0.05; **P < 0.01; ***P ≤ 0.001; ****P ≤ 0.0001.

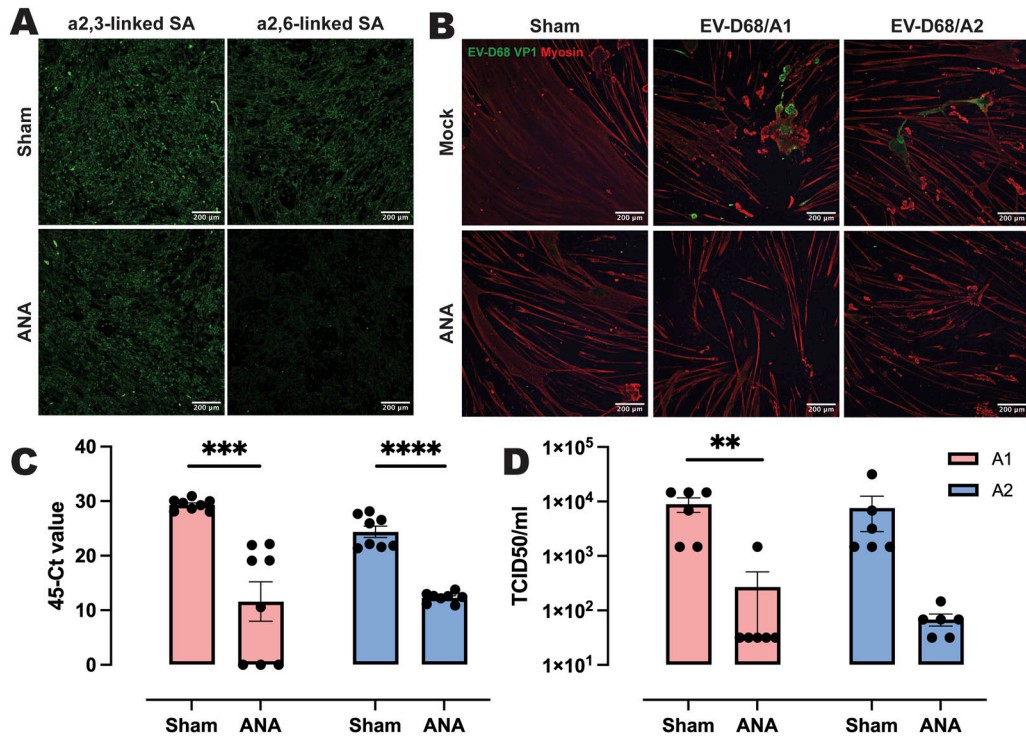

**Figure 2. Neuraminidase treatment of hiPSC-derived myotubes significantly reduced EV-D68 infection.**
**(A)** Expression of α2,3-linked and α2,6-linked SAs on hiPSC-derived myotubes with sham or *Arthrobacter ureafaciens* neuraminidase (ANA) treatment at 0 hpi. **(B)** Presence of EV-D68 VP1$^+$ cells in sham-treated, but not in ANA-treated, hiPSC-derived myotubes inoculated with EV-D68/A1 or EV-D68/A2 at MOI 0.1 at 24 hpi. Green: EV-D68 VP1; red: myosin. **(C, D)** Intracellular viral RNA (C) or infectious viruses in the supernatant (D) of hiPSC-derived myotubes treated with sham or ANA before inoculation with EV-D68/A1 (pink) or EV-D68/A2 (blue). Cells and supernatants were collected at 24 hpi. Three experiments were performed with two technical replicates. For detection of intracellular viral RNA, one measurement was repeated because of incomplete results from the previous run and the results from both measurements were included. Error bars indicate SEM. Statistical analysis was performed using an unpaired *t* test. **$P < 0.01$; ***$P ≤ 0.001$; ****$P ≤ 0.0001$.

In our human-derived model, EV-D68 is readily myotropic and myovirulent, and these attributes do not seem to be clade-dependent. Nonetheless, we observed differences in virulence among the tested viruses and, to a lesser extent, differences in replication efficiency among the included donors, indicating that virus-to-virus and host-to-host differences can still be contributing factors to different symptom progression and clinical outcomes in vivo. Which viral and host factors are responsible for these differences remains a subject of future investigations.

The ability to infect and damage skeletal muscle cells is not unique among enteroviruses. Virus replication in skeletal muscles has been shown to contribute to muscle damage and paralysis in vivo for enterovirus A71 (EV-A71) (26, 27, 28), echoviruses 30 (29) and 11 (30), coxsackieviruses A10 and A6 (31, 32, 33, 34), and poliovirus (35, 36). Human skeletal muscles have also been shown to harbour enteroviruses in patients with chronic inflammatory muscle disease, chronic fatigue syndrome, and polymyositis, suggesting that the enteroviruses can cause chronic muscular diseases (37, 38, 39, 40). In the case of EV-D68, it is not known whether virus replication in skeletal muscles contributes to muscle weakness and paralysis in humans, although the virus has been shown to cause paralysis without CNS involvement in mouse models (11). Intramuscular inoculation of neonatal mice with EV-D68 resulted in severe damage and apoptosis in the murine

muscle tissue related to viral replication (41). Whether the damage that we observed in our 2D and 3D models is also due to apoptosis remains to be investigated. It is also not known how severe the muscle damage is in EV-D68 patients with symptoms ranging from muscle ache, muscle weakness to complete paralysis. The depletion of satellite cells in our model suggests that muscle regeneration is hampered upon EV-D68 infection, which may contribute to different sequelae in patients. Full strength recovery of affected limbs is rare, but not impossible, in patients with EV-D68–associated AFM. In a surveillance study in the USA, only 5% of the AFM patients recovered fully (42). Whether satellite cell depletion affects the prospect of full recovery in patients remains to be determined. Further investigations are required to fully understand the role of skeletal muscles in the pathogenesis of EV-D68–associated muscle weakness and paralysis in humans.

Skeletal muscles are also an important bridge to the nervous system for various enteroviruses, because infection in the skeletal muscles can impair the neuromuscular junction (26, 27, 28) or facilitate virus spread to neurons through retrograde axonal transport (36). Muscle damage has also been shown to increase the efficiency of transport of poliovirus to the CNS (35). It is thus likely that muscle infection and its subsequent damage contribute to the spread of EV-D68 to the CNS. It is also important to note that there may be different abilities among EV-D68 isolates to enter the CNS

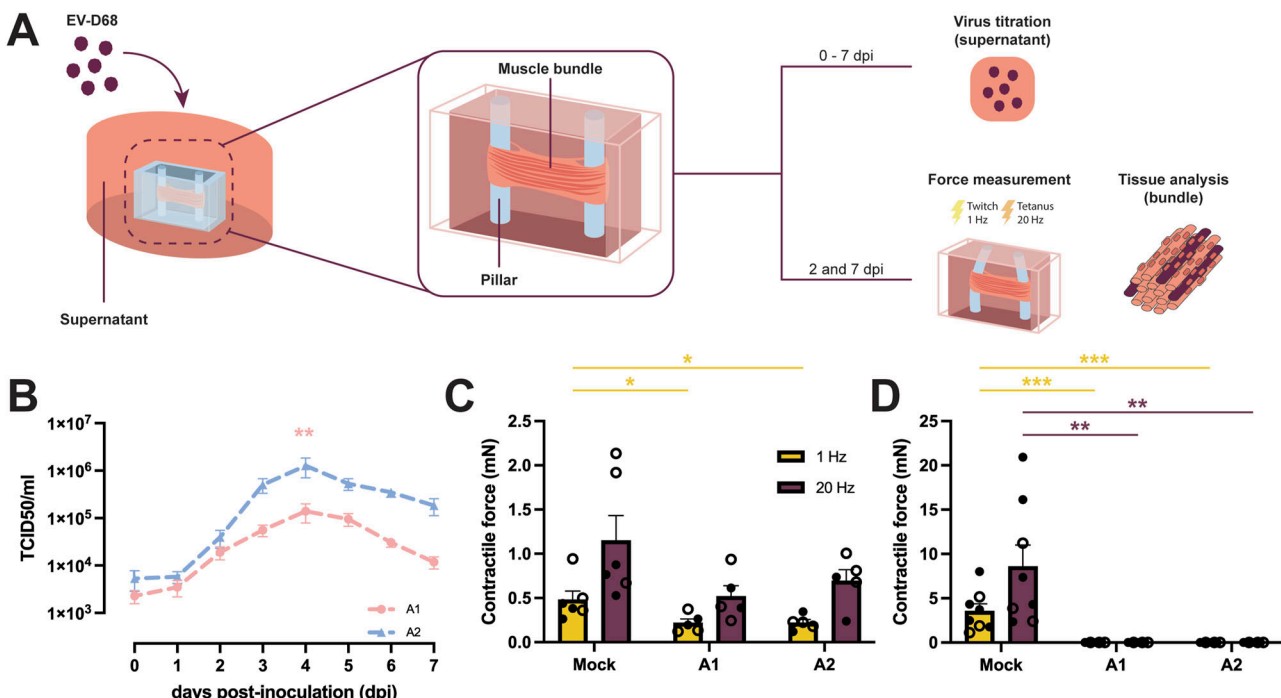

**Figure 3. Inoculation of 3D tissue-engineered skeletal muscles (TESMs) with EV-D68 resulted in infection of the tissues and loss of contractile force.**
**(A)** Schematic overview of the experimental workflow. The 3D TESMs were inoculated with EV-D68/A1 or EV-D68/A2 at 5 d post-differentiation (dpi). Supernatants were collected from 0 up to 7 dpi. Twitch and tetanic contractile forces were measured at 2 and 7 dpi, after which the 3D TESMs were collected for tissue analyses; **(B)** Virus titres from supernatants of 3D TESMs (n = 2 donors) inoculated with EV-D68/A1 (pink) and EV-D68/A2 (blue) and collected from 0 to 7 dpi. The asterisk indicates a significant difference between EV-D68/A1 and EV-D68/A2 at the peak of viral replication (4 dpi). Statistical analysis was performed using the Mann–Whitney test. **(C, D)** Average absolute contractile force of mock- and EV-D68–inoculated 3D TESMs after stimulation with 1 and 20 Hz at (C) 2 dpi and (D) 7 dpi. The asterisk indicates a significant difference of twitch contractile force (1 Hz; yellow) and tetanic force (20 Hz; purple) between mock- and virus-inoculated 3D TESMs at 2 or 7 dpi. Two experiments were performed with three technical replicates. Statistical analysis was performed using an unpaired $t$ test. *$P < 0.05$; **$P < 0.01$; ***$P ≤ 0.001$.

by retrograde transport from the muscles, because inoculation of $MAVS^{-/-}$ mice with paralytic and non-paralytic EV-D68 isolates resulted in similar virus replication in the muscles, but different neurological invasion (13). In addition, inoculation route seems to be a determinant for paralysis, as intramuscular inoculation leads to a higher frequency of paralysis in mice than intracranial inoculation, depending on the viral strain and the titre of inoculum (14, 43).

The pathogenesis of EV-D68–associated AFM, including the route of entry into the nervous system, remains an enigma. It is possible that the virus replicates first in skeletal muscles, causes damage of the neuromuscular junction, and subsequently infects the peripheral and central nervous system. In hiPSC-derived motor neurons grown in a microfluidic chamber, EV-D68 is transported retrogradely in spinal motor neurons (16). The second possible route of neuromuscular invasion of EV-D68 is through direct infection of the central and peripheral nervous system, which is followed by denervation of the motor neurons. In either scenario, virus infection will eventually lead to muscle weakness or paralysis. It is thus imperative to understand the neuromuscular invasion route of EV-D68 to be able to design a better, more accurately targeted treatment. It is important to take into account that our 3D TESM model does not fully replicate the in vivo situation and that virus inoculation may be more efficient in this model than in vivo. Nonetheless, our 2D

and 3D models still offer a powerful new tool to study viral myotropism and myovirulence.

Although we observed that EV-D68 infection of skeletal muscle cells is largely facilitated by the presence of sialic acids in our model, EV-D68 entry to neuronal cells can be independent of sialic acids (16, 44) and can instead be facilitated by intercellular adhesion molecule (ICAM)-5 and major facilitator superfamily domain–containing 6 (MFSD6). These molecules are highly expressed on neuronal cells (45, 46) and have been reported as EV-D68 entry receptors and can potentially be used by the virus to invade the CNS (47, 48). According to the Human Protein Atlas, human skeletal muscles do not express ICAM-5 (49) and only express MFSD6 at the moderate level (50). It will thus be of great interest to study the different entry mechanisms used to infect the skeletal muscles and further invade the motor neurons.

Altogether, we have demonstrated that hiPSC-derived 2D and 3D skeletal muscles are susceptible and permissive for EV-D68 isolates. The infection results in cellular damage and subsequently functional impairment of skeletal muscle tissues. Considering the importance of skeletal muscles in the pathogenesis of enterovirus infection and, especially in the case of EV-D68, in the development of paralysis, more in-depth studies on human skeletal muscles are required to design better prevention, interference, and recovery strategies that can benefit the patients.

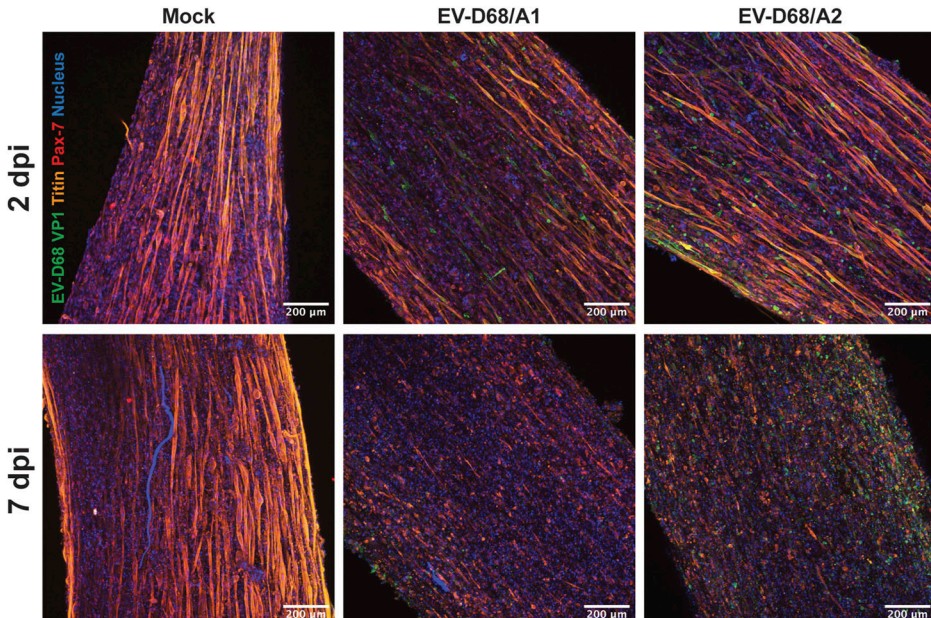

**Figure 4.** **Representative maximum intensity stacked images of 3D tissue-engineered skeletal muscles inoculated with EV-D68 at 2 and 7 dpi.**
EV-D68/A1 and EV-D68/A2 infection of 3D tissue-engineered skeletal muscles led to loss of titin⁺ skeletal muscle cells at 7 dpi. Green: EV-D68 VP1; red: Pax7; orange: titin; blue: nucleus.

# Materials and Methods

## Ethical statement

The iPSC lines used in this study were obtained in compliance with ethical guidelines, including appropriate donor consent and institutional approvals. Ethical approval for the LUMCi011-A line (Donor 1 for the 2D model) (51) was granted by the Research Subjects Review Board at the University of Rochester (RSRB reference: 00059324). Ethical approval for the 80RD60 line (Donor 2 for the 2D model) (52) was approved by the Erasmus MC review board. The HPSI0114i-kolf_3 (Donor 3 for the 2D model and Donor 1 for the 3D model) and AICS-TTN (Donor 2 for the 3D model) lines were purchased commercially from the Human Induced Pluripotent Stem Cell Initiative (HipSci) and Allen Institute for Cell Science (AICS) biobanks, respectively, where full informed consent was obtained from donors for research purposes. The HipSci general consent form was approved by the National Research Ethics Service (NRES) Committee East of England (REC reference: 15/EE/0049). The AICS consent form was approved by the University of California, San Francisco Institutional Review Board (IRB reference: 10-02521).

## Cells

Rhabdomyosarcoma (RD) cells (ATCC) were maintained in DMEM (Capricorn Scientific), with 10% heat-inactivated FBS, 100 IU/ml of penicillin, 100 µg/ml of streptomycin, and 2 mM L-glutamine.

hiPSC-derived myogenic progenitors (MPs) were generated and cultured according to the previously published protocol (21, 53). Briefly, the cells were seeded into a 10-cm tissue culture dish coated with ECM extract (prediluted 1:200 in DMEM with high glucose, E1270; Sigma-Aldrich) 20 min before cell seeding in proliferation medium consisting of DMEM with high glucose, 10% heat-inactivated FBS, 100 IU/ml of penicillin, 100 µg/ml of streptomycin, 2 mM L-glutamine, and 100 ng/ml basic FGF2 (bFGF2; PeproTech). For cell dissociation, the cells were incubated at 37°C and 5% $CO_2$ in the presence of TrypLE Express (Thermo Fisher Scientific). The cells were passaged twice in a 10-cm tissue culture dish for maintenance and expansion, after which the cells were seeded in a 48-well plate for differentiation into 2D myotubes.

## Viruses

EV-D68 strains included in this study were obtained from the National Institute of Public Health and the Environment (RIVM), Bilthoven, the Netherlands. The original clinical specimens, from which these viruses were isolated, were collected by RIVM for diagnostics purposes. The viruses were isolated from RD cells (ATCC) at 33°C at RIVM from respiratory samples from patients with EV-D68–associated respiratory disease. Virus stocks were generated in RD cells at 33°C and 5% $CO_2$. The viruses included in this study with year of isolation and accession number are as follows: subclade A1 (4311200821, 2012, accession number MN954536), subclade A2 (4311801122, 2018, accession code MN726791), and subclade B2 (B2/039; 4311201039, 2012, accession number MN954539).

## Whole-genome sequencing primer design

Primers were designed using PrimalScheme (https://primalscheme.com) based on 84 available EV-D68 whole-genome sequences from GenBank, which were aligned in BioEdit. The primers yielded seven PCR amplicons of ~1,000 nucleotides (Fig S9) with an overlapping region of ~200 nucleotides.

## RNA isolation, cDNA synthesis, and multiplex PCR

Viral RNA was collected using High Pure RNA Isolation Kit (Roche) according to the manufacturer's instructions and eluted in 100 $\mu$l elution buffer. The hybridisation step of the cDNA was generated using the SuperScript IV First-Strand cDNA synthesis kit (Thermo Fisher Scientific) in a total volume of 20 $\mu$l, containing 5 $\mu$l of eluted RNA, 10 mM of dNTPs, 20 U of RNase inhibitor, and 10 $\mu$M of the odd and even reverse primer pools. The cDNA reaction was performed at 42°C for 5 min, 50°C for 10 min, and 80°C for 10 min.

cDNA was subsequently amplified in two reactions (odd and even primer pools) using a multiplex PCR with a total volume of 50 $\mu$l containing 4 $\mu$l of undiluted cDNA, 5 $\mu$l of 10 PFU DNA polymerase buffer, 12.5 mM of dNTPs, 1 $\mu$l of PFU DNA polymerase (Agilent), and forward and reverse primers (Table S1). Multiplex PCR was performed at 95°C for 3 min; 40 cycles of 95°C for 20 s, 55°C for 30 s, and 72°C for 1 min; followed by 72°C for 3 min.

## PCR purification and Nanopore sequencing

Multiplex PCR products (5 $\mu$l) were analysed on a 1% agarose gel. The remaining products from both primer pools were combined and purified using Agencourt AMPure XP beads (Beckman Coulter). The product concentrations were measured with a Qubit dsDNA High Sensitivity assay kit (Thermo Fisher Scientific) on a Qubit fluorometer (Thermo Fisher Scientific); 65 ng of DNA was prepared per sample for library preparation. A maximum of 96 purified samples were barcode-labelled using the Native Barcoding kit 96 (SQK-NBD-114-96) according to the manufacturer's instructions and sequenced using a R10.4.1 MinION flowcell (Oxford Nanopore Technologies) on a GridION Mk1 (Oxford Nanopore Technologies) for 16 h.

## Sequence data analysis

Obtained sequence data were demultiplexed using default Nanopore software. The demultiplexed sequence data were analysed using CLC Genomics Workbench version 24.0.3.0 (QIAGEN). After trimming the 30-nucleotide primer sequences from the ends of the reads, sequence reads of the virus stocks were mapped against their respective reference sequences to identify cell culture–adaptive mutations. A consensus sequence was extracted with a minimum threshold of 25 read coverage per nucleotide. The original data were uploaded into the European Nucleotide Archive with the project accession code PRJEB93871 (secondary accession code ERP176747), with run file accession codes ERR15316058–ERR15316063 for EV-D68/A1 (2012); ERR15316064–ERR15316070 for EV-D68/A2 (2018); and ERR15316071–ERR15316076 for EV-D68/B2 (2012).

## EV-D68 reference sequences

Whole-genome sequences of the clinical EV-D68 isolates from subclades A1 (4311200821, 2012, accession number MN954536), A2 (4311400720, 2014, accession number MN954537), and B2 (B2/039; 4311201039, 2012, accession number MN954539) were included in the whole-genome analysis as reference strains. For the clinical

EV-D68/A2 isolate (4311801122, 2018, accession number MN726791), only the *VP1* region gene sequence was available in GenBank. Therefore, the whole-genome sequence of another EV-D68/A2 isolated in 2014 (4311400720, accession number MN954537) was used as its reference sequence.

## Generation of 2D myotubes

MPs (~100,000 cells/well; n = 3 donors) were seeded into a 48-well plate coated with ECM in the presence of proliferation medium. When cells reached ~90% confluency, the medium was changed to differentiation medium, consisting of DMEM with 1% penicillin and streptomycin, 2 mM L-glutamine, 1% Knockout Serum Replacement (KOSR; Thermo Fisher Scientific), and 1% Insulin/Transferrin/Selenium 100X (ITS-X; Thermo Fisher Scientific). A new medium was added after 48 h. The cells were fused into multinucleated myotubes during the culture period and were ready for inoculation 3 d after switching to differentiation medium.

## Generation of 3D TESMs

Polydimethylsiloxane (PDMS)-based 3D culture chambers with flexible pillars were fabricated according to the previously published method (54). 3D-TESMs (n = 2 donors) contained $6 \times 10^5$ cells per tissue. Hydrogel mixture (50 $\mu$l per tissue) contained 10% bovine fibrinogen (final concentration 2 mg/ml; Sigma-Aldrich), 20% Matrigel Growth Factor Reduced (Corning), and MPs and was prepared on ice. Cross-linking of fibrinogen was initiated by adding 0.5 units/ml of bovine thrombin (Sigma-Aldrich), after which the mixture was directly pipetted inside the PDMS chambers. 3D TESMs were incubated for 30 min at 37°C before the addition of 3D TESM growth medium, consisting of MP growth medium supplemented with 1.5 mg/ml 6-aminocaproic acid (6-ACA) (Sigma-Aldrich). After 2 d, differentiation was induced by switching medium to 2D myotube differentiation medium, supplemented with 2 mg/ml 6-ACA. Every 48 h, half of the medium was refreshed. 3D TESMs were cultured at 37°C with 5% $CO_2$.

## Inoculation of 2D myotubes and 3D TESMs with EV-D68

Myotubes were inoculated with EV-D68 at MOIs of 0.01, 0.1, and 1 for subclades A1 and A2, unless specified otherwise, and an MOI of 0.1 for subclade B2. The MOIs were calculated based on the number of MPs seeded per well in a 48-well plate. After 1 h, the inoculum was removed and the myotubes were washed three times with DMEM before supplementation with differentiation medium. At 0, 24, 48, and 72 hpi, the cells were collected for detection of viral capsid protein VP1 (by immunofluorescence assay) and intracellular viral RNA (by quantitative RT–PCR [RT–qPCR]), and the supernatants were collected for virus titration.

After 5 d of differentiation, 3D TESMs were inoculated with EV-D68/A1 and EV-D68/A2 at an MOI of 0.1. Similar to the 2D myotubes, the MOI was calculated based on the number of MPs seeded for the generation of 3D TESMs. After 1 h, the inoculum was removed, and the TESMs were washed three times with DMEM before supplementation with TESM differentiation medium. Supernatants were collected daily from 0 to 7 days post-inoculation (dpi) for virus

titration. After force measurement, TESMs were collected for immunofluorescence and immunohistochemistry assays. 3D TESMs that broke before or after the inoculation were excluded from the experiment.

### 3D TESM contractile force measurement

For electrical stimulations and force measurements, an Arduino Uno Rev3 equipped with an Adafruit Motor Shield V2 was used. Carbon plate electrodes were oriented parallel to the major axis of 3D TESMs. Stimulations were performed at a frequency of 1 or 20 Hz with 2.45 V and a duty cycle of 10%. Displacement of pillars was recorded with a DCC3240M camera (Thorlabs) at 60 frames per second and analysed with ImageJ for displacement. Pillar position of 3D TESMs was measured via images from the back of the pillar. Forces (expressed in N) were calculated with the following equation:

$$F = \left[\frac{6E\pi r^4}{4a^2(3L - a)}\right]\delta,$$

where E = Young's modulus of PDMS; r = radius of the pillar; a = height of the tissue on the pillar; L = length of the pillar; $\delta$ = displacement.

### Quantitative RT–PCR

To quantify viral RNA, a RT–qPCR was performed using 1X TaqMan Fast Virus 1-step Master Mix (Applied Biosystems). Primers and probes (Table S2), which bind to the 5′ untranslated region of EV-D68, used in this assay have been described previously (55). RT–qPCR was performed at 50°C for 5 min, 95°C for 20 s, followed by 45 cycles at 95°C for 3 s, and 60°C for 31 s using the ABI 7500 system (Thermo Fisher Scientific).

### Immunofluorescence assay

Myotubes were fixed and permeabilised using the Cytofix/Cytoperm Fixation and Permeabilisation kit (BD Biosciences) according to the manufacturer's instructions. The cells were incubated with a primary antibody mix, consisting of mouse anti-myosin heavy chain (Developmental Studies Hybridoma Bank; MF20) and rabbit anti-EV-D68 VP1 (GTX132313; GeneTex). After a washing step, the cells were incubated with a secondary antibody mix, consisting of donkey anti-mouse IgG-Alexa Fluor 555 (A31570; Invitrogen), donkey anti-rabbit IgG-Alexa Fluor 488 (A21206; Invitrogen), and Hoechst 33342.

TESMs were fixed with 4% PFA and permeabilised with 0.5% Triton X, 3% BSA, 0.1% Tween-20 diluted in PBS, each step for 1 h at room temperature. The bundles were incubated overnight at 4°C with a primary antibody mix, consisting of mouse anti-titin (IgM; Developmental Studies Hybridoma Bank; 9D10), mouse anti-paired box protein 7 (IgG1; Developmental Studies Hybridoma Bank; Pax7), and rabbit anti-EV-D68 VP1, in staining solution (0.1% Triton X, 0.1% BSA, and 0.1% Tween-20 in PBS). After a washing step, the bundles were incubated with the same secondary antibody mix as described above, with the addition of anti-mouse IgM-Alexa Fluor 647

(A-21238; Invitrogen), for 1 h at room temperature. For detection of proliferating satellite cells, the bundles were incubated with primary antibody mix that consists of mouse anti-Pax7 and rat anti-Ki67 (#740008T, clone SolA15; Thermo Fisher Scientific). After a washing step, the bundles were incubated with a secondary antibody mix consisting of donkey anti-mouse IgG-Alexa Fluor 555 and anti-rat-IgG2a-Alexa Fluor 647 (ab172333; Abcam). Fluorescence signals were detected using an inverted confocal laser scanning microscope LSM 700 (Zeiss), with EC Plan-Neofluar 10×/0.3 M27 objective lens. The microscope has four lasers: 405, 488, 555, and 639; and is equipped with bandwidth filters BP 490-555 (for 488), BP 505-600 (for 555), and LP 615 (for 639). Images were processed with FIJI software (ImageJ).

### Immunohistochemistry assay

Immunohistochemistry was performed on formalin-fixed, paraffin-embedded 3D TESMs that were collected at 2 and 7 dpi (n = 2 donors per time point). Formalin-fixed, paraffin-embedded 3D TESMs were sectioned at 3 $\mu$m, deparaffinised, and rehydrated before antigen retrieval in citrate buffer (10 mM, pH = 6.0) with heat induction. Sections were incubated with a primary antibody consisting of rabbit anti-EV-D68 VP1 or mouse anti-Ki67 (clone MIB-1; GA62661-2; DAKO) overnight at 4°C before incubation with secondary antibodies consisting of polyclonal goat anti-rabbit IgG (P04481-2; DAKO) or goat anti-mouse IgG1 (1071-05; Southern Biotech) conjugated with horseradish peroxidase. Peroxidase activity was revealed by incubating slides in 3-amino-9-ethylcarbazole (AEC) (Sigma-Aldrich) for 10 min, resulting in a bright red precipitate, and followed by a counterstaining with haematoxylin. Images were taken with Zen software.

### Virus titration

Virus titres in supernatants were assessed by end-point titrations in RD cells and were expressed as median tissue culture infectious dose (TCID$_{50}$/ml). In brief, 10-fold serial dilutions of a virus stock were prepared in triplicate and inoculated onto a monolayer of RD cells in DMEM with 10% FBS, 100 IU/ml of penicillin, 100 $\mu$g/ml of streptomycin, and 2 mM L-glutamine. The inoculated plates were incubated at 33°C in 5% CO$_2$. Cytopathic effect (CPE) was observed and recorded at day 5, and virus titres were determined using the Spearman–Kärber method (56).

### Removal of cell surface sialic acids on 2D myotubes

Myotubes (n = 1 donor) were incubated with 50 mU/ml *A. ureafaciens* neuraminidase (Roche) in serum-free medium for 2 h at 37°C in 5% CO$_2$ before inoculation with EV-D68/A1 and EV-D68/A2 at MOI 0.1. Removal of $\alpha$(2,3)-linked and $\alpha$(2,6)-linked sialic acids on the cell surface was verified by staining with biotinylated *Maackia amurensis* lectin (MAL) I (5 $\mu$g/ml; B-1265-1; Vector Laboratories) and fluorescein-labelled *Sambucus nigra* lectin (SNA) (5 $\mu$g/ml; BA-6802-1; EY Laboratories), respectively. Biotin was detected using a streptavidin-conjugated Alexa Fluor 488 (5 $\mu$g/ml; S11223; Thermo Fisher Scientific). Virus and mock inoculations in non–enzymatic-treated cells were included as positive and

negative infection controls, respectively. The removal of sialic acids was confirmed by detection of fluorescence signals using an inverted confocal laser scanning microscope. Three experiments were performed with two technical replicates.

### Statistical analysis

For each donor (n = 3), the 2D myotube experiments were performed three times with two technical replicates, unless specified otherwise. One-way ANOVA was used to analyse statistical differences in intracellular RNA levels and virus titres from cells from different donors inoculated with different viruses. For 3D TESMs from each donor (n = 2), two experiments were performed with three technical replicates. Some 3D TESMs that broke before or after virus inoculation were excluded from the analysis. The Mann–Whitney test was used to analyse the statistical difference in virus titre at 4 dpi from 3D TESMs inoculated with different viruses. An unpaired $t$ test was used to analyse the statistical difference in twitch and maximum tetanic contractile forces between mock- and virus-inoculated 3D TESMs. Statistical analyses were performed using GraphPad Prism 10.

### Citation software

Citations were organised using Endnote 20.6.

# Data Availability

All authors declare that all data generated and analysed in this study are included in this publication and its Supplementary Information files. The accession numbers of virus reference sequences are listed in the Materials and Methods section. The original data of whole-genome sequencing are available in the European Nucleotide Archive with the project accession code and the run file accession codes listed in the Materials and Methods section.

# Supplementary Information

# Acknowledgements

The authors would like to thank Adam Meijer for providing the virus isolates and Anjali Bholasing, Wilbert Vlot, Syriam Sooksawasdi Na Ayudhya, Feline Benavides, Nuder Nower Nizam, Vera Mols, and Keshia Kroh for technical assistance. This work was funded by a fellowship to DvR from the Netherlands Organisation for Scientific Research (VIDI contract 91718308). The collaboration project on hiPSC-derived myogenic progenitors, 2D myotubes, and 3D TESMs is co-funded by the PPP Allowance made available by Health~Holland, TopSector LifeSciences and Health, to the Prinses Beatrix Spierfonds to stimulate public–private partnerships (project numbers LSHM17075, LSHM19015, and LSHM20011) and by the National Growthfund project NXTGEN Hightech. The funders had no role in study design, data collection and analysis, decision to publish, or preparation of the article.

## Author Contributions

BM Laksono: conceptualisation, formal analysis, investigation, methodology, and writing—original draft, review, and editing.
AJ Bergsma: conceptualisation, formal analysis, investigation, methodology, and writing—review and editing.
A Iuliano: conceptualisation, formal analysis, investigation, methodology, and writing—review and editing.
DY Veldhoen: investigation.
S van Nieuwkoop: formal analysis, investigation, methodology, and writing—review and editing.
M Boter: resources, investigation, and writing—review and editing.
L Leijten: resources and writing—review and editing.
L Bauer: resources, formal analysis, and writing—review and editing.
BB Oude Munnink: supervision, methodology, and writing—review and editing.
WWMP Pijnappel: conceptualisation, resources, supervision, methodology, project administration, and writing—review and editing.
D van Riel: conceptualisation, resources, supervision, funding acquisition, methodology, project administration, and writing—original draft, review, and editing.

## Conflict of Interest Statement

The authors declare that they have no conflict of interest.

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
