## [Reviewer comments · Life Science Alliance]

Life Science Alliance

Elucidating the role of human skeletal muscles in the pathogenesis of enterovirus D68 infection

Brigitta Laksono, Atze Bergsma, Alessandro Iuliano, Dominique Veldhoen, Stefan van Nieuwkoop, Marjan Boter, Lonneke Leijten, Lisa Bauer, Bas Oude Munnink, W.W.M. Pijnappel, and Debby van Riel

DOI: <https://doi.org/10.26508/lsa.202503372>

Corresponding author(s): *Debby van Riel, Erasmus MC*

Review Timeline:	Submission Date:	2025-04-29
	Editorial Decision:	2025-06-18
	Revision Received:	2025-07-22
	Editorial Decision:	2025-08-11
	Revision Received:	2025-08-21
	Accepted:	2025-08-25

Scientific Editor: Sarita Hebbar

Transaction Report:

June 17, 2025

Re: Life Science Alliance manuscript #LSA-2025-03372

Dr. Debby van Riel
Erasmus MC
Dr Molewaterplein 40
Rotterdam, Zuid Holland 3015 GD
Netherlands

Dear Dr. van Riel,

Thank you for submitting your manuscript entitled "Elucidating the role of human skeletal muscles in the pathogenesis of enterovirus D68 infection" to Life Science Alliance. The manuscript was assessed by three expert reviewers, whose comments are appended to this letter.

All three reviewers commented that the work is interesting. However we agree with the reviewers that the manuscript needs to be revised before publication at LSA. A revised manuscript must include:

1. A clarification on the potential role of muscle injury in pathogenesis versus AFM -related weakness (Reviewer 1, paragraph 2)
2. A statement on the choice of clades and MOIs used in this study (Reviewer 1, specific comment 4 and Reviewer 2, points 2,3)
3. Better explanation OR additional data to support:
 - descriptions in Figure 1A and Figure 4 (Reviewer 3, comments 1 and 4)
 - comparisons of effect on myofibers between EV-D68 A1, A2 and B2 (Reviewer 3, comment 3)
 - interpret the contractile strength changes (Reviewer 1, point 7)
4. All clarifications requested by reviewers connected to details of methods and reagents used in this study (Reviewer 2, several points)

In line with the overall recommendations, we invite you to submit a revised manuscript addressing the Reviewer comments. When submitting the revision, please include a letter addressing the reviewers' comments point by point. While a rebuttal must respond to all points in some form, additional data to resolve these points (other than ones indicated above) may not be required.

Thank you for this interesting contribution to Life Science Alliance. We are looking forward to receiving your revised manuscript.

Sincerely,

Sarita Hebbbar, PhD
Scientific Editor
Life Science Alliance
<http://www.lsajournal.org>

B. MANUSCRIPT ORGANIZATION AND FORMATTING:

Reviewer #1 (Comments to the Authors (Required)):

Authors present a detailed characterization of EV-D68 infection by viruses belonging to three subclades (A1, A2, B2) in iPSC derived human skeletal muscle myotubes grown in 2 and 3D culture.

Weakness is the cardinal feature of the subset of EV-D68 infected children who develop acute flaccid myelitis (AM). The evidence is in this reviewer's opinion overwhelming that the dominant cause of this is viral infection of and associated injury and death of spinal cord motor neurons. This motor neuron focus is apparent in the unique human pathological material (see Vogt ref. 9) but also consistent with a large body of available spinal MRI and electrophysiological (EMG) data. There is very little to no evidence in humans (in contrast to mice) for the presence of substantial myositis or muscle injury in humans or that such injury plays essentially any role at all in the EV-D68 associated AFM-related weakness. In fairness however I want to note that in terms of pathogenesis the route by which EV-D68 reaches and infects motor neurons has not been established- and murine studies absolutely indicate that virus can spread from skeletal muscle to Spinal cord motor neurons to produce a murine AFM like illness. It is fair to conjecture that viremia and subsequent localization of virus in skeletal muscle with subsequent retrograde transport to motor neurons could be a pathway of infection (and again as noted this clearly occurs in mice in terms of the muscle to motor neuron path and demonstrated retrograde spread in motor neurons-multiple refs cited by authors (e.g. 14, 39, 40 etc.). I just think the authors need to be clearer that it is the potential role in pathogenesis not the direct muscle injury accounts for AFM weakness component- as this is otherwise misleading?

Some specific comments:

(1) Commendation to the authors for the careful work validating sequences of the viruses and absence of TC acquired mutations. This is more critical here as the cells virus is grown in (RD) are "myosarcoma" and such TC mutations might have affected their pathogenesis in human myotubes.

(2) Lines 68 et seq. -please correct to be clearer that "complete recovery" in fact is vanishingly rare in EVD68-AFM. (perhaps the authors inadvertently lumped A71 AFM cases in their thinking as in those recovery is much higher and can be more complete?)

(3) lines 78 et seq. not all the refs cited (11-14) were in fact in IFN alpha/beta receptor deficient mice- e.g I don't believe 13 or 14 were??

(4) Surprised that the B3 subclade was not included given its current predominance in most areas? In addition given recent decline in AFM after 2018(as opposed to respiratory EV-D68 infections which have re-emerged post-COVID) it would be interesting to know if the myotube predilection had changed for the most recent subclades?

(5) Fig 2 data does strongly suggest sialic acids (neuraminidase sensitive) play a key but not absolute role in viral infection (and presumed entry) in human myotubes. The authors could clarify better in discussion the role of sialic acid in neuronal entry as this seems/remains controversial (see also DM Brown et al mBio, '18: doi.org/10.1128/mBio.01954-18 suggesting some strains were

more dependent than others on sialic acid). The authors might comment on which other putative receptors are present on myotubes-e.g. ICAM5/telencephalon, MFSD6?

(7) The muscle function studies are unique- it would help readers to understand how to interpret the contractile strength changes (mN) as these will likely be unfamiliar to most. Given lack/lesser growth of B2 subclade- I would have been interested to see that as a "control" for non-specific injury to muscle independent of replication.

(8) There is some data suggesting EV-D68 can trigger apoptosis/caspase activation in muscle in mice-(see Frost J in J Virol 2023; doi.org/10.1128/jvi.00156-23) do the authors have a comment on mechanism of the muscle injury they see- is it apoptotic?

Reviewer #2 (Comments to the Authors (Required)):

The authors' present evidence that EV-D68 infects skeletal muscles and interferes with contractile force and interrupts muscle regeneration using both 2D and 3D in vitro culture models. The authors used several clinically-relevant EV-D68 clades (A1, A2 and B2). This study builds on knowledge about how EV-D68 can cause muscle dysfunction without necessarily infecting motor neurons. Below are some suggestions to help improve the manuscript:

1. Please provide an ethics statement for this work.
2. Why did the authors chose A1, A2 and B2 clades of EV-D68 for this work? Adding additional justification would be helpful for the reader since other EV-D68 clades were more prominent in being linked with AFM. For example, clade B1 was found to be more associated with AFM in 2014.
3. The choice of viral clades and MOIs used in these experiments is not well explained and additional justification would be helpful for the reader. The authors performed a viral titration of all 3 clades by infecting cells with MOI 0.01 to 1. However, why the authors then chose MOI=0.1 for performing additional work with clade B if better CPE was observed with MOI=1 with A1 is unclear.
4. Figure 1A - there appears to be clear presence of CPE in the B2 viral isolate used in the infections but then authors used A1 and A2 to do the subsequent functional work in their 3D cultures. Justification why they chose A1 and A2 over B2 would be helpful.
5. Can the authors further comment why they saw a slight difference in viral kinetics between the clinical isolates in their experiments (Figure 1B and C)? Were the sequences from each donor different? This was unclear. Also, can the authors comment as to which isolate they finally chose to further study as this was unclear in the manuscript (i.e. from which donor).
6. Methods - please include the original NGS data in a publically available database. The NGS results plus consensus for viral isolates obtained. This was not provided in the manuscript.
7. Line 341 - authors trimmed 30- nucleotide sequences from the ends of reads. What tool did they use to remove these sequences and was the full primer removed or just 30 nucleotides from the ends? Please clarify.
8. Line 431-433 - please include incubation conditions for the secondary antibody mix (time and temp)
9. Please include scale bar for supplemental figure S6

Reviewer #3 (Comments to the Authors (Required)):

The manuscript, "Elucidating the role of human skeletal muscles in the pathogenesis of enterovirus D68 infection" by Laksono et al, discusses the Enterovirus D68 (EV-D68), a respiratory virus with extra-respiratory effects such as acute flaccid myelitis. The authors generate myofibers by induced pluripotent stem cell (iPSC) differentiation into 2D and 3D models, and show that EV-D68/A1, A2 and B2 (belonging to the A and B virus clades) can infect myofibers, leading to cell death. By treating with neuraminidase which cleaves sialic acid, they show that sialic acids facilitate myofiber infection by EV-D68. Infection also led to reduced myofiber contractility and tetanic force. The Authors also claim that virus infection leads to reduced number of Pax7+ muscle progenitors and muscle sarcomere components such as Titin.

Overall, this manuscript puts forward a set of observations, describing how the EV-D68 virus could infect the skeletal muscle and induce muscle-associated pathology. The observations are interesting and need some more clarity, especially with respect to the effect of the virus on differentiation, which would add to the study. Therefore, I recommend that the manuscript be considered again after revision, if the Authors satisfactorily address the detailed comments below.

Specific comments:

1. Lines 116-120: "rounding, detachment and eventually cell death" - please point to examples of each of these in Figure 1A.
2. In Figure 1 legend, for panel A, "Representative images of skeletal muscles inoculated...." is inappropriate to use; these are cultured muscle fibers and should be referred to as such. The same applies to the legend of panels B-C and the entire manuscript.
3. Lines 133-134: "We observed that the progression of EV-D68/B2 CPE, signified by the increased number of dead and infected cells at 24 hpi, resembled that of EV-D68/A1 instead of A2". It is unclear why the Authors feel this way; based on the

images in panel A, EV-D68/B2 seems to have the strongest effect on myofibers, with hardly any myofibers seen at 48 and 72 hpi, compared to EV-D68/A1 and EV-D68/A2. The graphs in panels B-C also do not show anything that makes EV-D68/B2 seem more similar to EV-D68/A2 than EV-D68/A1. Therefore, the Authors need to clearly substantiate their claim in lines 133-134 with the reasoning behind it and data to support it.

4. Figure 4 is strange, where it is hard to distinguish the Pax7 and Titin staining. A clearer description of what is shown in this figure is needed. The effect on Titin, which labels the Z-disc of the sarcomeres is not mentioned in the results or discussion.

5. The effect of EV-D68 on myogenic differentiation should be studied and included in the manuscript. This could have a fundamental effect on muscle function following virus infection, that the Authors are characterizing.

Dear Dr. van Riel,

Thank you for submitting your manuscript entitled "Elucidating the role of human skeletal muscles in the pathogenesis of enterovirus D68 infection" to Life Science Alliance. The manuscript was assessed by three expert reviewers, whose comments are appended to this letter.

All three reviewers commented that the work is interesting. However we agree with the reviewers that the manuscript needs to be revised before publication at LSA. A revised manuscript must include:

1. A clarification on the potential role of muscle injury in pathogenesis versus AFM - related weakness (Reviewer 1, paragraph 2)
2. A statement on the choice of clades and MOIs used in this study (Reviewer 1, specific comment 4 and Reviewer 2, points 2,3)
3. Better explanation OR additional data to support:
 - descriptions in Figure 1A and Figure 4 (Reviewer 3, comments 1 and 4)
 - comparisons of effect on myofibers between EV-D68 A1, A2 and B2 (Reviewer 3, comment 3)
 - interpret the contractile strength changes (Reviewer 1, point 7)
4. All clarifications requested by reviewers connected to details of methods and reagents used in this study (Reviewer 2, several points)

In line with the overall recommendations, we invite you to submit a revised manuscript addressing the Reviewer comments. When submitting the revision, please include a letter addressing the reviewers' comments point by point. While a rebuttal must respond to all points in some form, additional data to resolve these points (other than ones indicated above) may not be required.

To upload the revised version of your manuscript, please log in to your

account: <https://lsa.msubmit.net/cgi-bin/main.plex>

Thank you for this interesting contribution to Life Science Alliance. We are looking forward to receiving your revised manuscript.

Sincerely,

Sarita Hebbar, PhD
Scientific Editor
Life Science Alliance
<http://www.lsajournal.org>

- A letter addressing the reviewers' comments point by point.
- An editable version of the final text (.DOC or .DOCX) is needed for copyediting (no PDFs).
- High-resolution figure, supplementary figure and video files uploaded as individual files: See our detailed guidelines for preparing your production-ready images, <https://www.life-science-alliance.org/authors>
- Summary blurb (enter in submission system): A short text summarizing in a single sentence the study (max. 200 characters including spaces). This text is used in conjunction with the titles of papers, hence should be informative and complementary to the title and running title. It should describe the context and significance of the findings for a general readership; it should be written in the present tense and refer to the work in the third person. Author names should not be mentioned.
- By submitting a revision, you attest that you are aware of our payment policies found here: <https://www.life-science-alliance.org/copyright-license-fee>

B. MANUSCRIPT ORGANIZATION AND FORMATTING:

Reviewer #1 (Comments to the Authors (Required)):

Authors present a detailed characterization of EV-D68 infection by viruses belonging to three subclades (A1, A2, B2) in iPSC derived human skeletal muscle myotubes grown in 2 and 3D culture.

Weakness is the cardinal feature of the subset of EV-D68 infected children who develop acute flaccid myelitis (AM). The evidence is in this reviewer's opinion overwhelming that the dominant cause of this is viral infection of and associated injury and death of spinal cord motor neurons. This motor neuron focus is apparent in the unique human pathological material (see Vogt ref. 9) but also consistent with a large body of available spinal MRI and electrophysiological (EMG) data. There is very little to no evidence in humans (in contrast to mice) for the presence of substantial myositis or muscle injury in humans or that such injury plays essentially any role at all in the EV-D68 associated AFM-related weakness. In fairness however I want to note that in terms of pathogenesis the route by which EV-D68 reaches and infects motor neurons has not been established- and murine studies absolutely indicate that virus can spread from skeletal muscle to Spinal cord motor neurons to produce a murine AFM like illness. It is fair to conjecture that viremia and subsequent localization of virus in skeletal muscle with subsequent retrograde transport to motor neurons could be a pathway of infection (and again as noted this clearly occurs in mice in terms of the muscle to motor neuron path and demonstrated retrograde spread in motor neurons-multiple refs cited by authors (e.g. 14, 39, 40 etc.). I just think the authors need to be clearer that it is the potential role in pathogenesis not the direct muscle injury accounts for AFM weakness component- as this is otherwise misleading?

Authors: we agree with the reviewer that the potential role of skeletal muscles in the pathogenesis of EV-D68 infection is as the bridge to the motor neurons. However, we cannot dismiss the possibility that skeletal muscles does not play a role in EV-D68-associated muscle weakness. Events between the respiratory infection and motor neuron infection and loss are poorly understood, but several patient cases have described the development of muscle weakness prior to complete limb

paralysis and AFM (e.g. Rodesch *et al.*, 2024, <https://doi.org/10.1159/000535316>; Kreuter *et al.*, 2011, <https://doi.org/10.5858/2010-0174-CR.1>; Chan *et al.*, 2021, <https://doi.org/10.12809/hkmj208931>). In this very early development of muscle weakness, we cannot exclude the possibility that this muscle weakness is caused by EV-D68 replication in skeletal muscles, which could subsequently spread further into the motor neurons via the neuromuscular junctions.

As also explored in our Discussion (lines 236-239): *Human skeletal muscles have also been shown to harbour enteroviruses in patients with chronic inflammatory muscle disease, chronic fatigue syndrome and polymyositis, suggesting that the enteroviruses can cause chronic muscular diseases (37-40)*; Unfortunately, these available studies did not further type the enteroviruses in these patients, but these reports argue in favour of a possibility of skeletal muscle infection by enteroviruses, which potentially include EV-D68.

We have also adjusted our Discussion to tone down our speculation (lines 239-241): *In the case of EV-D68, it is not known if virus replication in skeletal muscles contributes to muscle weakness and paralysis in humans, although the virus has been shown to cause paralysis without CNS involvement in mouse models (11)*.

Some specific comments:

(1) Commendation to the authors for the careful work validating sequences of the viruses and absence of TC acquired mutations. This is more critical here as the cells virus is grown in (RD) are "myosarcoma" and such TC mutations might have affected their pathogenesis in human myotubes.

Authors: we thank the reviewer for their commendation.

(2) Lines 68 et seq. -please correct to be clearer that "complete recovery" in fact is vanishingly rare in EVD68-AFM. (perhaps the authors inadvertently lumped A71 AFM cases in their thinking as in those recovery is much higher and can be more complete?)

Authors: while rare, complete recovery has been reported in some AFM cases (e.g. Martin *et al.*, <https://doi.org/10.1212/WNL.0000000000004081>, Chong *et al.*, <https://doi.org/10.1016/j.pediatrneurol.2020.11.019>). However, we agree that these complete recovery cases are rare and we corrected our sentence into (lines 67-69): "*Clinical outcome of AFM differs widely, ranging from the more common life-long muscle weakness and atrophy to the very rare cases of complete recovery.*"

(3) lines 78 et seq. not all the refs cited (11-14) were in fact in IFN alpha/beta receptor deficient mice- e.g I don't believe 13 or 14 were??

Authors: we have adjusted the sentence and the references accordingly (lines 78-80): *Studies in interferon- (IFN-) α/β receptor-deficient (11, 12) and neonatal mice (13, 14) have shown that EV-D68 infects skeletal muscles and motor neurons, the latter ultimately leading to paralysis.*

(4) Surprised that the B3 subclade was not included given its current predominance in most areas? In addition given recent decline in AFM after 2018(as opposed to respiratory EV-D68 infections which have re-emerged post-COVID) it would be interesting to know if the myotube predilection had changed for the most recent subclades?

Authors: At the start of our study, we aim to include the currently circulating strains A2 and B3. Unfortunately, our B3 virus stock acquired several cell culture adaptive mutations while passaging, so we decided to exclude B3 from our study and focus on A2.

(5) Fig 2 data does strongly suggest sialic acids (neuraminidase sensitive) play a key but not absolute role in viral infection (and presumed entry) in human myotubes. The authors could clarify better in discussion the role of sialic acid in neuronal entry as this seems/remains controversial (see also DM Brown *et al* *mBio*, '18: doi.org/10.1128/mBio.01954-18 suggesting some strains were more dependent than others on sialic acid). The authors might comment on which other putative receptors are present on myotubes-e.g. ICAM5/telencephalon, MFSD6?

Authors: we have now adjusted our Discussion to implement the Reviewer's recommendation (lines 282-290): *While we observed that EV-D68 infection of skeletal muscle cells is facilitated by the presence of sialic acids in our model, EV-D68 entry to neuronal cells can be independent of sialic acids (16, 44) and can instead be facilitated by Intercellular Adhesion Molecule- (ICAM-)5 and Major Facilitator Superfamily Domain-containing 6 (MFSD6). These molecules are highly expressed on neuronal cells (45, 46) and have been reported as EV-D68 entry receptors and can potentially be utilised by the virus to invade the CNS (47, 48). According to the Human Protein Atlas, human skeletal muscles do not express ICAM-5 (49) and only express MFSD6 at moderate level (50). It will thus be of great interest to study the different entry mechanisms use to infect the skeletal muscles and further invade the motor neurons.*

(7) The muscle function studies are unique- it would help readers to understand how to interpret the contractile strength changes (mN) as these will likely be unfamiliar to most. Given lack/lesser growth of B2 subclade- I would have been interested to see that as a "control" for non-specific injury to muscle independent of replication.

Authors: we have added more explanation to help the interpretation of the contractile strength data in our Results (lines: 173 - 177): *Briefly, electrical stimulation were applied onto the TESMs and pillar displacement was tracked with high-speed video imaging as previously described (21). A functional TESM will show a single twitch contraction when stimulated with 1 Hz and reach a maximum tetanic contraction with a 20-Hz stimulus. These contractile forces can then be plotted into a graph (in mN).*

Overall, the RNA level and virus titre of B2 are similar to A1 and A2, with the exception of that at 72 hpi, where the viral titre is significantly lower compared to A1 (Reviewer Fig 1). However, based on CPE (Fig 1A) B2 triggers more CPE, suggesting comparable to A1. We therefore do not think that B2 would be a 'control for non-specific injury to muscles independent of replication'.

Reviewer Fig 1. Individual virus titres in A1-, A2- and B2-inoculated 2D myotube supernatant at different time points (0, 24, 48 and 72 hpi). Statistical difference was analysed using one-way ANOVA.

(8) There is some data suggesting EV-D68 can trigger apoptosis/caspase activation in muscle in mice-(see Frost J in J Virol 2023; doi.org/10.1128/jvi.00156-23) do the authors have a comment on mechanism of the muscle injury they see- is it apoptotic?

Authors: indeed it will be interesting to study the cause of myofiber death by EV-D68 in this model in more detail and whether it supports the observation in the mice of Frost *et al.* We adjusted our Discussion to address this (lines 241-244): *Intramuscular inoculation of neonatal mice with EV-D68 resulted in severe damage and apoptosis in the murine muscle tissue related to viral replication (41). Whether the damage that we observed in our 2D and 3D models is also due to apoptosis remains to be investigated.*

Reviewer #2 (Comments to the Authors (Required)):

The authors' present evidence that EV-D68 infects skeletal muscles and interferes with contractile force and interrupts muscle regeneration using both 2D and 3D in vitro culture models. The authors used several clinically-relevant EV-D68 clades (A1, A2 and B2). This study builds on knowledge about how EV-D68 can cause muscle dysfunction without necessarily infecting motor neurons. Below are some

suggestions to help improve the manuscript:

1. Please provide an ethics statement for this work.

Authors: We added an ethical statement in the Materials and Methods (lines 302-304): *The research project utilised myogenic progenitors derived from iPSCs of healthy individuals, for which ethical approval was obtained institutionally. This protocol ensures ethical oversight and adherence to institutional standards.*

2. Why did the authors chose A1, A2 and B2 clades of EV-D68 for this work? Adding additional justification would be helpful for the reader since other EV-D68 clades were more prominent in being linked with AFM. For example, clade B1 was found to be more associated with AFM in 2014.

Authors: although initially subclade B1 was thought to be associated with AFM, more subclades (including A2, previously known as D1) are also linked with AFM. More findings have also shown that EV-D68 neurotropism is not clade specific (Rosenfeld *et al.*, 2019; <https://doi.org/10.1128/mBio.02370-19>; Sooksawasdi Na Ayudhya *et al.*, 2020; <https://doi.org/10.1128/mSphere.00941-20>). Based on this, we decided to include strains from several clades, including A1 and A2 to compare "old" and "contemporary" strains within the same clade. We added this explanation in our Introduction (lines 87-89): *We included strains from clades A and B isolated before and after 2014, some of which are known to be associated with AFM (15) or can cause paralytic disease in mice (16).*

3. The choice of viral clades and MOIs used in these experiments is not well explained and additional justification would be helpful for the reader. The authors performed a viral titration of all 3 clades by infecting cells with MOI 0.01 to 1. However, why the authors then chose MOI=0.1 for performing additional work with clade B if better CPE was observed with MOI=1 with A1 is unclear.

Authors: also see our comment #2 above. Our initial studies with A1 and A2 included different MOIs, to define the best MOI for the rest of the study. While the CPE is better with MOI 1 for A1, we observed that the increase of viral RNA over time was

similar to those observed after inoculation with an MOI 0.1 or 0.01 (Fig S4A). With MOI 0.1, we observe CPE with both viruses and a clear increase of viral RNA and virus titres over time. Based on this observation we decided to continue the rest of the studies with MOI 0.1.

4. Figure 1A - there appears to be clear presence of CPE in the B2 viral isolate used in the infections but then authors used A1 and A2 to do the subsequent functional work in their 3D cultures. Justification why they chose A1 and A2 over B2 would be helpful.

Authors: see our comment #2 above.

5. Can the authors further comment why they saw a slight difference in viral kinetics between the clinical isolates in their experiments (Figure 1B and C)? Were the sequences from each donor different? This was unclear. Also, can the authors comment as to which isolate they finally chose to further study as this was unclear in the manuscript (i.e. from which donor).

Authors: please also see our response to Reviewer 1's comment #7. We sequenced the virus stocks (inoculum), but not the new infectious virus particles that were released into the supernatant, so we do not have any information regarding the sequences. To investigate whether there is donor-to-donor variation, we included 3 hiPSC donors. We observed significant differences as described in the Results (Fig 1C, circles), suggesting that there is a slight donor-to-donor variation that can affect EV-D68 replication.

6. Methods - please include the original NGS data in a publically available database. The NGS results plus consensus for viral isolates obtained. This was not provided in the manuscript.

Authors: per reviewer's suggestion, we have included the original NGS data into the European Nucleotide Archive. The project accession code is PRJEB93871 and we have included this too in the Materials and Methods (lines 368-372): *The original data was uploaded into the European Nucleotide Archive with project accession*

code PRJEB93871 (secondary accession code ERP176747), with run file accession codes ERR15316058 - ERR15316063 for EV-D68/A1 (2012); ERR15316064 - ERR15316070 for EV-D68/A2 (2018) and ERR15316071 - ERR15316076 for EV-D68/B2 (2012).

7. Line 341 - authors trimmed 30- nucleotide sequences from the ends of reads. What tool did they use to remove these sequences and was the full primer removed or just 30 nucleotides from the ends? Please clarify.

Authors: we trimmed 30-nt sequences using the CLS Genomics Workbench v24.0.4.0, which allows us to remove the entire primer binding sites.

8. Line 431-433 - please include incubation conditions for the secondary antibody mix (time and temp)

Authors: we have adjusted the Methods accordingly (lines 459-461): *After a washing step, the bundles were incubated with the same secondary antibody mix as described above, with the addition of anti-mouse IgM-Alexa Fluor647 (A-21238; Invitrogen), for 1 h at room temperature.*

9. Please include scale bar for supplemental figure S6

Authors: we have adjusted the figures accordingly.

Reviewer #3 (Comments to the Authors (Required)):

The manuscript, "Elucidating the role of human skeletal muscles in the pathogenesis of enterovirus D68 infection" by Laksono et al, discusses the Enterovirus D68 (EV-D68), a respiratory virus with extra-respiratory effects such as acute flaccid myelitis. The authors generate myofibers by induced pluripotent stem cell (iPSC) differentiation into 2D and 3D models, and show that EV-D68/A1, A2 and B2 (belonging to the A and B virus clades) can infect myofibers, leading to cell death. By treating with neuraminidase which cleaves sialic acid, they show that sialic acids facilitate myofiber infection by EV-D68. Infection also led to reduced myofiber

contractility and tetanic force. The Authors also claim that virus infection leads to reduced number of Pax7+ muscle progenitors and muscle sarcomere components such as Titin.

Overall, this manuscript puts forward a set of observations, describing how the EV-D68 virus could infect the skeletal muscle and induce muscle-associated pathology. The observations are interesting and need some more clarity, especially with respect to the effect of the virus on differentiation, which would add to the study. Therefore, I recommend that the manuscript be considered again after revision, if the Authors satisfactorily address the detailed comments below.

Specific comments:

1. Lines 116-120: "rounding, detachment and eventually cell death" - please point to examples of each of these in Figure 1A.

Authors: we have now indicated the examples with arrows and arrowhead in Fig 1A and adjusted the figure legend accordingly: *Arrowhead indicates a rounding cell; thick arrow indicates a detached cell; thin arrow indicates a dead cell.*

2. In Figure 1 legend, for panel A, "Representative images of skeletal muscles inoculated...." is inappropriate to use; these are cultured muscle fibers and should be referred to as such. The same applies to the legend of panels B-C and the entire manuscript.

Authors: we now refer to the cells or tissues as "myotubes" or "TESMs" and have made the changes accordingly throughout the whole manuscript.

3. Lines 133-134: "We observed that the progression of EV-D68/B2 CPE, signified by the increased number of dead and infected cells at 24 hpi, resembled that of EV-D68/A1 instead of A2". It is unclear why the Authors feel this way; based on the images in panel A, EV-D68/B2 seems to have the strongest effect on myofibers, with hardly any myofibers seen at 48 and 72 hpi, compared to EV-D68/A1 and EV-D68/A2. The graphs in panels B-C also do not show anything that makes EV-D68/B2

seem more similar to EV-D68/A2 than EV-D68/A1. Therefore, the Authors need to clearly substantiate their claim in lines 133-134 with the reasoning behind it and data to support it.

Authors: also see our respond to Reviewer 1's comment #7. Our claim is based on the timing of the appearance of CPE: A1 and B2 showed CPE (indicated by the presence of infected AND dead cells) already at 24 hpi, while A2 did not (we only observed the presence of infected cells but not many dead cells at 24 hpi). Additionally, based on their replication kinetics (Fig 1B-C), we cannot make the claim that B2 is more virulent nor has a more efficient replication compared to A1 and A2. If we plot the increase of viral RNA and titres relative to 0 hpi (arbitrary dotted line at 10^5 TCID50/ml is added to aid visualisation), B2 replicates similarly to A1 and A2 (Reviewer Fig 2 and 3). Based on this, we stand by our claim that B2 is similar to A1 in its ability to cause CPE and to A1 and A2 to replicate in our 2D model.

Reviewer Fig 2. The increase of intracellular viral RNA relative to 0 hpi for all viruses combined (top left), A1 only (top right), A2 (bottom left) and B2 (bottom right) from 3 different hiPSC donors.

Reviewer Fig 3. The increase of virus titres in the supernatant relative to 0 hpi for all viruses combined (top left), A1 only (top right), A2 (bottom left) and B2 (bottom right) from 3 different hiPSC donors.

4. Figure 4 is strange, where it is hard to distinguish the Pax7 and Titin staining. A clearer description of what is shown in this figure is needed. The effect on Titin, which labels the Z-disc of the sarcomeres is not mentioned in the results or discussion.

Authors: the staining of Pax7 indeed still gave some background signal when combined with titin staining, as can be seen in Fig 4. That is why we included S6-7 Figs to better see the Pax7 and Titin stainings in separate channels. We also performed an additional staining of Pax7 (without Titin staining) to further show the decrease of Pax7+ cells following the infection as shown in S8 Fig.

We did mention the effect of infection on titin in our Results (lines 189-190): *The infection progressed over time, resulting in destruction of the 3D TESMs structure, as marked by the loss of titin⁺ skeletal muscle cells at 7 dpi (Fig 4).*

5. The effect of EV-D68 on myogenic differentiation should be studied and included in the manuscript. This could have a fundamental effect on muscle function following virus infection, that the Authors are characterizing.

Authors: The cultures included in our study, both 2D and 3D, are fully differentiated. In addition, after the inoculation of EV-D68, the cultures did not survive, making further long-term observation impossible in this model.

August 11, 2025

RE: Life Science Alliance Manuscript #LSA-2025-03372R

Dr. Debby van Riel
Erasmus MC
Dr Molewaterplein 40
Rotterdam, Zuid Holland 3015 GD
Netherlands

Dear Dr. van Riel,

Thank you for submitting your revised manuscript entitled "Elucidating the role of human skeletal muscles in the pathogenesis of enterovirus D68 infection". Your revised manuscript was evaluated by the original reviewers, and they have acknowledged your efforts in addressing their concerns. Reviewers' comments are appended below.

We would be happy to publish your paper in Life Science Alliance pending final revisions necessary to meet our formatting guidelines.

- We recommend you to address the requirement from Reviewer 2 for (a) the details of the ethical statement (human material) reviewed by a ethics committee, and (b) reference for documentation for the donor materials that provided the EV-D68 samples.
- In the methods section:
 - Please provide a table with all primer sequences used in this study.
 - Please provide details for following antibody: anti-Pax7.
 - Please provide microscopy details: specify the objective (with NA) used and wavelengths for excitation and emission.
 - Please include details for image processing and citation for software used.
 - Please confirm that statistical significance between pairs (if applicable for A2) is shown in Figure 2D.
 - Please add a "Data Availability" statement providing details (the repository name and persistent identifier: DOI, accession number, or permanent URL) on manuscript data submitted to a public, open access repository.
 - Please add an Author Contributions section to your main manuscript text
 - Please add callouts for Figures S6A-B and S7A-B to your main manuscript text.
 - Please add the X and Bluesky handles of your host institute/organization, as well as your own and/or one of the authors in our system
 - Please be sure that the authorship listing and order is correct

A. FINAL FILES:

- An editable version of the final text (.DOC or .DOCX) is needed for copyediting (no PDFs).
- High-resolution figure, supplementary figure and video files uploaded as individual files: See our detailed guidelines for preparing your production-ready images, <https://www.life-science-alliance.org/authors>
- Summary blurb (enter in submission system): A short text summarizing in a single sentence the study (max. 200 characters

including spaces). This text is used in conjunction with the titles of papers, hence should be informative and complementary to the title. It should describe the context and significance of the findings for a general readership; it should be written in the present tense and refer to the work in the third person. Author names should not be mentioned.

B. MANUSCRIPT ORGANIZATION AND FORMATTING:

Sincerely,

Sarita Hebbar, PhD
Scientific Editor
Life Science Alliance
<http://www.lsajournal.org>

Reviewer #1 (Comments to the Authors (Required)):

I believe the authors have tried to reasonably respond to my comments and have no further issues.

Reviewer #2 (Comments to the Authors (Required)):

The authors have addressed majority of my concerns.

However, the ethical statement that the authors included in the revision should include the reference to documentation stating their use of human material was independently reviewed by an ethics committee. Furthermore, please also include reference to documentation for the donor materials that provided the EV-D68 samples as this was not in the revised statement.

Reviewer #3 (Comments to the Authors (Required)):

The Authors have satisfactorily addressed my comments and suggestions. I am fine for this manuscript to be accepted for publication.

Dear Dr. van Riel,

Thank you for submitting your revised manuscript entitled "Elucidating the role of human skeletal muscles in the pathogenesis of enterovirus D68 infection". Your revised manuscript was evaluated by the original reviewers, and they have acknowledged your efforts in addressing their concerns. Reviewers' comments are appended below.

We would be happy to publish your paper in Life Science Alliance pending final revisions necessary to meet our formatting guidelines.

-We recommend you to address the requirement from Reviewer 2 for (a) the details of the ethical statement (human material) reviewed by a ethics committee, and (b) reference for documentation for the donor materials that provided the EV-D68 samples.

-In the methods section:

--Please provide a table with all primer sequences used in this study.

Authors: All NGS primers are listed in Table S1 and the RT-PCR primers and probes are now listed in Table S2.

--Please provide details for following antibody: anti-Pax7.

Authors: The detail for anti-Pax7 is included this information in the Materials and Methods section (lines 468-469): *"mouse anti-paired box protein 7 (IgG1; Developmental Studies Hybridoma Bank; Pax7)..."*

--Please provide microscopy details: specify the objective (with NA) used and wavelengths for excitation and emission.

Authors: We have included the confocal microscopy details in the Materials and Methods section (lines 477-480): *... with EC Plan Neofluar 10x/0.3 M27 objective lens. The microscope has four lasers: 405, 488, 555 and 639; and is equipped with bandwidth filters BP 490-555 (for 488), BP 505-600 (for 555) and LP 615 (for 639).*

--Please include details for image processing and citation for software used.

Authors: The detail on image processing software is included in the Materials and Methods section (line 480): *"Images were processed with FIJI software (ImageJ)." We added a sub-section 'Citation software' (lines 529-530): "Citations were organised using Endnote 20.6."*

-Please confirm that statistical significance between pairs (if applicable for A2) is shown in Figure 2D.

Authors: We have performed statistical analyses on the data of Figure 2 and, despite the trend, did not observe any statistical significance between the A2 groups in Figure 2D.

-Please add a "Data Availability" statement providing details (the repository name and persistent identifier: DOI, accession number, or permanent URL) on manuscript data submitted to a public, open access repository.

Authors: We have added a Data Availability statement (lines 551-556):

"Data availability"

All authors declare that all data generated and analysed are included in this publication and its Supplementary Information files. Virus reference sequences accession numbers are listed in the Materials and Methods. The original data of whole genome sequencing are available in the European Nucleotide Archive with the project accession code and run file accession codes listed in the Materials and Methods."

-Please add an Author Contributions section to your main manuscript text

Authors: We have added an Author Contributions (lines 532-548).

-Please add callouts for Figures S6A-B and S7A-B to your main manuscript text.

Authors: The callouts for Figures S6 and S7 are included in the main manuscript (lines 194 and 200).

-Please add the X and Bluesky handles of your host institute/organization, as well as your own and/or one of the authors in our system

Authors: Our X handles are:

Brigitta M. Laksono: @bmlaksono
Atze J. Bergsma: @atzeb1
Alessandro Iuliano: @Ale_luliano92
Lonneke Leijten: @LonnekevNes
Lisa Bauer: @LisaBauerVirus
Bas Oude Munnink: @1986Bom
Debby van Riel: @DebbyvanRiel

Brigitta Laksono: @bmlaksono.bsky.social
Lisa Bauer: @lisabauervirus.bsky.social
Debby van Riel: @debbyvanriel.bsky.social

-Please be sure that the authorship listing and order is correct

Authors: We have confirmed that the authorship listing and order is correct.

LSA now encourages authors to provide a 30-60 second video where the study is briefly explained. We will use these videos on social media to promote the published paper and the presenting author (for examples, see <https://docs.google.com/document/d/1-UWCfbE4pGcDdcgzcmiuJI2XMBJnxKYegRvLLrLSo8s/edit?usp=sharing>). Corresponding or first-authors are welcome to submit the video. Please submit only one video per manuscript. The video can be emailed to contact@life-science-alliance.org

To upload the final version of your manuscript, please log in to your account: <https://lsa.msubmit.net/cgi-bin/main.plex>

A. FINAL FILES:

-- Summary blurb (enter in submission system): A short text summarizing in a single sentence the study (max. 200 characters including spaces). This text is used in conjunction with the titles of papers, hence should be informative and complementary to the title. It should describe the context and significance of the findings for a general

readership; it should be written in the present tense and refer to the work in the third person. Author names should not be mentioned.

B. MANUSCRIPT ORGANIZATION AND FORMATTING:

Sincerely,

Sarita Hebbbar, PhD
Scientific Editor
Life Science Alliance
<http://www.lsjournal.org>

Reviewer #1 (Comments to the Authors (Required)):

I believe the authors have tried to reasonably respond to my comments and have no further issues.

Reviewer #2 (Comments to the Authors (Required)):

The authors have addressed majority of my concerns.

However, the ethical statement that the authors included in the revision should include the reference to documentation stating their use of human material was independently reviewed by an ethics committee. Furthermore, please also include reference to documentation for the donor materials that provided the EV-D68 samples as this was not in the revised statement.

Authors: We have included a more detailed Ethical Statement (lines 302-314): *The iPSC lines used in this study were obtained in compliance with ethical guidelines, including appropriate donor consents and institutional approvals. Ethical approval for the LUMCi011-A line (Donor 1 for the 2D model) (51) was granted by the Research Subjects Review Board at the University of Rochester (RSRB reference: 00059324). Ethical approval for the 80RD60 line (Donor 2 for the 2D model) (52) was approved by the Erasmus MC review board. The HPSI0114i-*

kolf_3 (Donor 3 for the 2D model and Donor 1 for the 3D model) and AICS-TTN (Donor 2 for the 3D model) lines were purchased commercially from the Human Induced Pluripotent Stem Cell Initiative (HipSci) and Allen Institute for Cell Science (AICS) biobanks, respectively, where full informed consent was obtained from donors for research purposes. The HipSci general consent form was approved by the National Research Ethics Service (NRES) Committee East of England (REC reference: 15/EE/0049). The AICS consent form was approved by the University of California San Francisco Institutional Review Board (IRB reference: 10-02521).

Regarding the donor material: We do not have any information about the donors as we received virus stocks from the National Institute of Public Health and the Environment (RIVM), Bilthoven, the Netherlands. The original clinical samples were collected by for diagnostics purposes from which viruses were isolated. To make this clear for the readers, we have adjusted our Materials and Methods (lines 333-336): *"EV-D68 strains included in this were obtained from the National Institute of Public Health and the Environment (RIVM), Bilthoven, the Netherlands."*

Reviewer #3 (Comments to the Authors (Required)):

The Authors have satisfactorily addressed my comments and suggestions. I am fine for this manuscript to be accepted for publication.

August 25, 2025

RE: Life Science Alliance Manuscript #LSA-2025-03372RR

Dr. Debby van Riel
Erasmus MC
Dr Molewaterplein 40
Rotterdam, Zuid Holland 3015 GD
Netherlands

Dear Dr. van Riel,

Thank you for submitting your Research Article entitled "Elucidating the role of human skeletal muscles in the pathogenesis of enterovirus D68 infection". It is a pleasure to let you know that your manuscript is now accepted for publication in Life Science Alliance. Congratulations on this interesting work.

DISTRIBUTION OF MATERIALS:

Again, congratulations on a very nice paper. I hope you found the review process to be constructive and are pleased with how the manuscript was handled editorially. We look forward to future exciting submissions from your lab.

Sincerely,

Sarita Hebbar, PhD
Scientific Editor
Life Science Alliance
<http://www.lsajournal.org>